# High-level expression of the monomeric SARS-CoV-2 S protein RBD 320-537 in stably transfected CHO cells by the *EEF1A1*-based plasmid vector

Maria V. Sinegubova[1], Nadezhda A. Orlova[1]*, Sergey V. Kovnir[1], Lutsia K. Dayanova[2], Ivan I. Vorobiev[1,2]

1 Laboratory of Mammalian Cell Bioengineering, Institute of Bioengineering, Research Center of Biotechnology of the Russian Academy of Sciences, Moscow, Russia, 2 Laboratory of Glycoproteins Biotechnology, Institute of Bioorganic Chemistry of the Russian Academy of Sciences, Moscow, Russia

* nobiol@gmail.com

**Data Availability Statement:** All relevant data are within the manuscript and its Supporting information files. Plasmids are deposited in GenBank - p1.1-Tr2-eGFP [GenBank: MW187857];

## Abstract

The spike (S) protein is one of the three proteins forming the coronaviruses' viral envelope. The S protein of the Severe Acute Respiratory Syndrome Coronavirus 2 (SARS-CoV-2) has a spatial structure similar to the S proteins of other mammalian coronaviruses, except for a unique receptor-binding domain (RBD), which is a significant inducer of host immune response. Recombinant SARS-CoV-2 RBD is widely used as a highly specific minimal antigen for serological tests. Correct exposure of antigenic determinants has a significant impact on the accuracy of such tests–the antigen has to be correctly folded, contain no potentially antigenic non-vertebrate glycans, and, preferably, should have a glycosylation pattern similar to the native S protein. Based on the previously developed p1.1 vector, containing the regulatory sequences of the Eukaryotic translation elongation factor 1 alpha gene (*EEF1A1*) from Chinese hamster, we created two expression constructs encoding SARS-CoV-2 RBD with C-terminal c-myc and polyhistidine tags. RBDv1 contained a native viral signal peptide, RBDv2 –human tPA signal peptide. We transfected a CHO DG44 cell line, selected stably transfected cells, and performed a few rounds of methotrexate-driven amplification of the genetic cassette in the genome. For the RBDv2 variant, a high-yield clonal producer cell line was obtained. We developed a simple purification scheme that consistently yielded up to 30 mg of RBD protein per liter of the simple shake flask cell culture. Purified proteins were analyzed by polyacrylamide gel electrophoresis in reducing and non-reducing conditions and gel filtration; for RBDv2 protein, the monomeric form content exceeded 90% for several series. Deglycosylation with PNGase F and mass spectrometry confirmed the presence of N-glycosylation. The antigen produced by the described technique is suitable for serological tests and subunit vaccine studies.

p1.1-Tr2-RBDv1 [GenBank: MW187858]; pTM [GenBank: MW187855]; pTM-RBDv2 [GenBank: MW187856]; and available from Addgene: p1.1-Tr2-eGFP - #162782; pTM - #162783; pTM-RBDv2 - #162785.

**Funding:** The authors received no specific funding for this work.

**Competing interests:** MVS and LKD declare that they have no competing interests. SVK, NAO and IIV are inventors of the patent RU2488633, which covers the use of the p1.1 plasmid; a derivative of this plasmid is used in current research. The existence of the patent does not alter our adherence to PLOS ONE policies on sharing data and materials.

**Abbreviations:** ABC, ammonium bicarbonate solution; ACE2, angiotensin-converting enzyme 2; ACN, acetonitrile; CHO, Chinese hamster ovary; DHFR, dihydrofolatereductase; EEF1A1, Eukaryotic translation elongation factor-1 alpha gene; EMCV, encephalomyocarditis virus; HT, hypoxanthine - thymidine; IDA, iminodiacetic acid; IRES, internal ribosome entry site; MTX, methotrexate; NTA, nitrilotriacetic acid; ORF, open reading frame; PNGase F, peptide:N-glycosidase F; RBD, receptor-binding domain; SARS-CoV-2, Severe Acute Respiratory Syndrome Coronavirus 2; TFA, trifluoroacetic acid.

## Introduction

Humanity is faced with an unprecedented challenge—the severe acute respiratory syndrome coronavirus 2 (SARS-CoV-2), which causes a severe respiratory illness—coronavirus disease 2019 (COVID-19) pandemic. Countries were sent to lockdown; people could not make informed decisions about the possibility of social contacts; the need for diagnostic tests is very high. Existing tests for SARS-CoV-2 are reviewed in [1].

At the beginning of the pandemic, polymerase chain reaction (PCR) testing methods dominated since such test systems can be developed urgently, soon after the emergence of a new virus in the population. Serological testing makes it possible to reliably determine whether a person is infected with the SARS-CoV-2, even in the absence of disease symptoms, or long after the event of infection. Serologic tests are also needed to detect convalescent plasma of therapeutic interest and assess emerging vaccines' effectiveness.

The immunodominant antigen of SARS-CoV-2 is the receptor-binding domain (RBD) of the spike (S) protein [2]. Another antigen widely used for diagnostics—the nucleoprotein (N) protein—combines high sensitivity and low specificity. Cases are described for SARS-CoV when the results of testing with N-protein were clarified using two subunits of S-protein [3]. S-protein and its fragments can be used not only for routine testing for anti-SARS-CoV-2 antibodies presence but also for mapping of neutralizing antibodies epitopes, as was done for SARS-CoV [4] and for affordable surrogate virus neutralization tests, based, for example, on antibody-mediated blockage of ACE2 –S interaction [5].

The S protein plays a crucial role in receptor recognition, cell membrane fusion, internalization of viruses, and their exit from the endosomes [6]. The S protein is co-translationally incorporated into the rough endoplasmic reticulum and is glycosylated by N-linked glycans. Interacting with the M and E proteins S protein trimer is transported to the virus's assembly site. S protein is required for cell entry but not necessary for virus assembly [7].

During their intracellular processing, S proteins of many types of coronaviruses, including SARS-CoV-2 and the Middle East respiratory syndrome-coronavirus (MERS-CoV), but not SARS-CoV, undergo partial proteolytic degradation at the furin signal protease recognition site with the formation of two subunits S1 and S2. The S protein homotrimer binds to the ACE2 dimer; a detailed study of this interaction is available [8].

A glycan shield is formed by N-linked glycans on the S protein surface, which is likely to help viral immune escape. In a comparative study of genome-wide sequencing data of natural isolates of SARS-CoV-2 [9] for the detected 228 variants of the S protein, all 22 potential N-glycosylation sites within the S protein's ectodomain were completely conserved, which confirms the importance of each of these sites for maintaining the integrity of the S protein oligosaccharide envelope.

Not all of these 22 potential N-glycosylation sites are occupied, for S1 and S2 subunits, obtained separately by the transiently transfected Human Embryonic Kidney cells 293 (HEK293) [10] N-glycosylation events were experimentally confirmed only for 17 out of 22 sites. At least one O-glycosylation site was experimentally found inside the RBD-domain area of the S1 subunit with the mucin-like oligosaccharide structures. Non-vertebrate cells can be used to produce the S protein or its fragments; in this case, N-glycans are present mostly in the form of bulky high mannose or paucimannose structures, excessively blocking the interaction of antibodies with the folded S protein [11]. Computational modeling of the glycan shield, performed for the HEK293-derived S protein, revealed that around 40% of the protein's surface is effectively shielded from IgG antibodies [12].

The use of full-length S protein for routine serological testing is nearly impossible due to its insolubility, caused by the transmembrane domain. An artificial trimer of its ectodomains was

used in serological tests [13]; however, such complex protein cannot be obtained in large quantities in mammalian cells. It is generally believed that the SARS-CoV-2 S protein receptor-binding domain is a minimal proteinaceous antigen, adequately resembling the immunogenicity of the whole spike protein. This domain contains only two occupied N-glycosylation sites [10] and 1–2 occupied O-glycosylation sites. It does not contribute to the trimer formation, and its surface is mostly unshielded.

Isolated RBD's of the S proteins of beta-coronaviruses were produced in various expression systems. Bacterial expression of the RBD from MERS-CoV produced no soluble target protein; refolding attempts also were unsuccessful [14]. Budding yeasts *Pichia pastoris* were the suitable host for the secretion of MERS-CoV RBD with at least two (from three) N-linked glycosylation sites present. Similar data were obtained for the RBD from SARS-CoV virus–removal of all N-glycosylation sites resulted in the sharp drop of protein secretion rate in the *P. pastoris* yeast; in the case of full RBD domain (residues 318–536), secretion of the un-glycosylated target protein was stopped completely [15]. It can be assumed that N-glycosylation is essential for correct folding of the RBD.

The SARS-CoV-2 S protein RBD was also completely insoluble upon *Escherichia coli* expression; solubilized inclusion bodies were unreactive even on blotting [16]. Hyperglycosylated yeast-derived SARS-CoV-2 RBD was obtained in reasonable quantities (50 mg/L in bioreactor culture) and successfully used for mice immunization [17]. Unfortunately, yeast-derived glycosylated proteins are not suitable for serological testing since they contain immunogenic glycans. Similarly, SARS-CoV-2 RBD produced in the *Nicotiana benthamiana* plant contains non-vertebrate N-glycans, potentially reactive with human antibodies [18].

Most of the early publications on the SARS-CoV-2 S protein and its RBD domain production in mammalian cells describe the transient transfection of HEK293 cells [10, 19] and purification of small protein lots in a very short time. For example, D. Stadlbauer [20] reports more than 20 mg/L titers in transiently transfected HEK-293 cells. The scalability of transiently transfected cell lines cultivation is still questionable; the gram quantities of RBD for large-scale serological testing can be produced only by stably transfected cell lines.

Previously we have developed the plasmid vector p1.1, containing large fragments of non-coding DNA from the *EEF1A1* gene of the Chinese hamster and fragment of the Epstein-Barr virus long terminal repeat concatemer [21] and employed it for unusually high-level expression of various proteins in Chinese hamster ovary (CHO) cells, including blood clotting factors VIII [22], IX [23], and heterodimeric follicle-stimulating hormone [24]. CHO cells were successfully used for transient SARS-CoV RBD expression at 10 mg/L secretion level [25]. We have proposed that SARS-CoV-2 RBD, suitable for in vitro diagnostics use, can be expressed in large quantities by stably transfected CHO cells, bearing the EEF1A1-based plasmid.

## Materials and methods

### Molecular cloning

p1.1-Tr2-RBDv1 construction. The RBD 319–541 coding sequence was synthesized according to [13], synthetic gene SARS_CoV_2RBD_his [GenBank: MT380724.1]. The DNA fragment encoding the RBDv1 open reading frame (ORF) with Kozak consensus sequence and C-terminal c-myc and 6xHis tags were obtained by PCR using primers AD-COV-AbsF and AD-RBD-myc6HNheR (listed in Table 1) and Tersus polymerase mix (Evrogen, Moscow, Russia). Synthetic oligos, PCR reagents, Plasmid Miniprep Purification kit, PCR Clean-Up System were from Evrogen. The PCR product was restricted using *Abs*I (Sibenzyme, Novosibirsk, Russia) and *Nhe*I (Fermentas, Vilnius, Lithuania) enzymes and inserted into p1.1-Tr2 vector instead of eGFP (p1.1-Tr2-eGFP [GenBank: MW187857] is available from Addgene, plasmid

**Table 1. Primers used for molecular cloning.**

| Primers for RBDv1 cloning, restriction sites are underlined | |
| --- | --- |
| AD-COV-AbsF | AA**CCTCGAGG**CCGCCACCATGTTCATGCCTTCTT |
| AD-RBD-myc6HNheR | **GCTAGC**CTAATGGTGATGGTGATGATGACCGGTATGCATATTCAGATCCTCTTCTGAGATGAGTTTTTGTTCGAAGTTCACGCATTTGTT |
| **Primers for pTM construction, sticky ends of annealed pairs are underlined** | |
| AS-Myc6H-AbsF | **TCGA**GGCCGCCACCATGGATGCAATGAAGAGAGGGCTCTGCTGTGTGCTGCTGCTGTGTGGAGCAGTCTTCGTCTCGGCTA |
| AS-Myc6H-AbsR | **GCGC**TAGCCGAGACGAAGACTGCTCCACACAGCAGCAGCACACAGCAGAGCCCTCTCTTCATTGCATCCATGGTGGCGGCC |
| AS-Myc6H-SpeF | **GCGC**TACCGGTGAACAAAAACTGATCAGCGAAGAGGATCTGTCTGCAGGCGGTCATCACCATCACCATCACCATCA |
| AS-Myc6H-SpeR | **CTAG**TGATGGTGATGGTGATGGTGATGACCGCCTGCAGACAGATCCTCTTCGCTGATCAGTTTTTGTTCACCGGTA |
| **Primers for RBDv2 cloning, restriction sites are underlined** | |
| AD-SFR2-NheF | **GCTAGC**GTGCAGCCCACCGAATCC |
| AD-SFR2-XmaR | **CCCGGG**TTTGTTCTTCACGAGATTGGT |
| **Sequencing primers** | |
| SQ-5CH6-F | GCCGCTGCTTCCTGTGAC |
| IRESA rev | AGGTTTCCGGGCCCTCACATTG |
| SQ-MycH-R | GATGACCGCCTGCAGAC |

#162782). The resulting plasmid p1.1-Tr2-RBDv1 was sequenced using SQ-5CH6-F and IRESA rev primers (Table 1).

PTM vector construction. A polylinker was constructed from synthetic oligos listed in Table 1. Oligos were 5'phosphorylated by manufacturer, annealed in pairs (AS-Myc6H-AbsF with AS-Myc6H-AbsR; AS-Myc6H-SpeF with AS-Myc6H-SpeR) in equimolar concentrations in a thermocycler (heated 95°C for 3 minutes, and gradually cooled down -2° C per cycle for 36 cycles), ligated by T4 and cloned to p1.1-Tr2-eGFP restricted by *Abs*I-*Nhe*I in place of the eGFP insert. Polylinker had *Abs*I-*Spe*I sticky overhangs, so *Nhe*I site in the vector backbone was destroyed–this provided an opportunity to use a unique *Nhe*I site inside the new polylinker. The Resulting pTM vector was sequenced as described above, available from Addgene, plasmid #162783.

pTM-RBDv2 construction. RBD ORF was amplified using adaptor primers AD-SFR2-NheF and AD-SFR2-XmaR restricted by *Nhe*I and *Xma*I (Sibenzyme, Novosibirsk, Russia) and cloned into pTM vector, restricted by *Nhe*I and *AsiG*I (Sibenzyme, Novosibirsk, Russia). The resulting construct was sequenced using SQ-5CH6-F and SQ-MycH-R primers. pTM-RBDv2 is available from Addgene, plasmid #162785.

Plasmids for cell transfections were purified by the Plasmid Midiprep kit (Evrogen, Moscow, Russia) and concentrated by ethanol precipitation in sterile conditions.

**Quantitative PCR, PCR.** The transgene copy number in the CHO genome was determined by the quantitative real-time-PCR (qPCR) as described in [21, 23]. Serial dilutions of p1.1-eGFP [21] or pGem-Rab1 plasmids were used for calibration curves generation. The weight of one CHO haploid genome was taken as 3 pg, according to [26]. Genomic DNA (gDNA) was purified by The Wizard® SV Genomic DNA Purification System (Promega, USA), diluted to 10ng/µl, 50 ng of gDNA used for one PCR reaction. For genome insert quantification primers RT-ID-F (5'-GCCACAAGATCTGCCACCATG-3') and RT-ID-R (5'-GTAGGTCTCCGTTCTTGCCAATC-3') were used, RT-Rab1-F (5'-GAGTCCTACGCTAATGTGAAAC-3') and RT-Rab1-R (5'-TTCCTTGGCTGTGGTGTTG-3') were used for normalization. Primers and qPCRmix-HS SYBR reaction mixture (Evrogen) and iCycler iQ thermocycler (Bio-Rad, USA) were used. Calculations of threshold cycles, calibration curves, PCR efficiency and copy numbers were made by the iCycler Iq4 program.

## Cell culture

Chinese hamster ovary DG-44 cells (Thermo Fischer Scientific) were cultured in the ProCHO 5 medium (Lonza, Switzerland), supplemented by 4 mM glutamine, 4 mM alanyl-glutamine and hypoxanthine-thymidine supplement (HT) (PanEco, Moscow, Russia). Cells were grown as a suspension culture in sterile 125 ml Erlenmeyer flasks with vented caps, routinely passaged 3 to 4 days with centrifugation (300 g, 5 min) and seeding density 3–4×10exp5 cells/ml.

## Stably transfected cell lines generation and genomic amplification of target gene

The 50–80 µg of each plasmid were precipitated by the addition of 96% ethanol and 3M sodium acetate, washed with 70% ethanol, dried, and resuspended in 100 µl of sterile R-buffer, Neon transfection kit (Thermo Fischer Scientific). The cells were passaged 24 h before transfection. Thirty millions cells per transfection were pelleted by centrifugation, washed once with DPBS (Paneco), and resuspended in 100 µl of the plasmid solution described. Five micrograms of reporter plasmid pEGFP-C2 (Takara Bio Inc., Kusatsu, Shiga, Japan), coding the green fluorescent protein, was added for each transfection. Cells were nucleofected with Neon Transfection system (Thermo Fischer Scientific) at 1700 V, single pulse, 20 ms, 100 µl transfection tip and placed in 30 ml warm ProCHO 5 medium supplemented with 1xHT and 8 mM glutamine and cultured for 48 h in 125 ml Erlenmeyer shaking flask. The transfection efficiency was assessed 48 h post-transfection as the proportion of eGFP-positive live cells. Transfection efficiency was 9% in the case of RBDv1 and 24% in the case of RBDv2. The medium was changed for the selective one, lacking HT and containing 200 nM of methotrexate (MTX, Sandoz, Holzkirchen, Germany). Cells were passaged every 3–4 days until the viability was restored to 85% (20–25 days). After generating a stably transfected cell line up to two rounds of target gene amplification were performed by increasing MTX concentration from 200 nM to 2 µM and 8 µM. Cells were passaged every 3–4 days until the viability restored to 85%; the typical time for establishing genome-amplified cultures was 12–21 days.

## Preparative cell cultures

Seeding cell culture was grown in 125 ml Erlenmeyer shake flasks with 30 ml of Lonza Pro-CHO 5 medium, supplemented with 4 mM glutamine, 4 mM alanyl-glutamine and 2–8 µM MTX until cell concentration exceeds 1–1.5 ×10 exp6 cells/ml. Cell suspension was transferred to four 250 ml Erlenmeyer flasks, each containing 60 ml of culture medium, and grown to the same cell density. The entire cell suspension was transferred to a single 2 L Erlenmeyer flask with 1 L culture medium, final seeding density 3–4×10exp5 cells/ml. Cells were cultured for three days, on the fourth day of culture, daily glucose measurements were started. Glucose concentration in the cell supernatant was measured by the Accutrend Plus system (Roche, Switzerland); if glucose level was below 20 mM, it was added up to 50 mM as the sterile 45% solution. The culture in 2 L flask was grown for 6 to 8 days until the cell viability, measured by trypan blue exclusion, dropped below 50%.

## Clonal cell line generation

The clonal cell line was obtained by the limiting dilution method from the cell population, cultured in 8 µM MTX. Methotrexate was omitted in the culture medium for two 3 d passage before cloning. Cells were additionally split by 1:1 dilution 24 hours before the cloning procedure. Cells were diluted in EXCELL-CHO (Merck, Germany) culture medium supplemented with 4 mM glutamine, 4 mM alanyl-glutamine, HT and 10% of untransfected CHO DG 44

conditioned medium resulting in seeding density 0.5 cell/well, and the suspension was seeded into 96-well plates (200 μl/well). Plates were left undisturbed for 14 days at 37˚C, 5% CO2 atmosphere. Wells with single colonies were screened by microscopy; well grown colonies were detached by pipetting and transferred to the wells of 12-well plate, containing 4 ml of the EXCELL-CHO, supplemented as described above and grown for 7 days undisturbed. Product titer was measured by ELISA, as described below, 6 wells with highest RBDv2 titer were used for further cultivation. Best-producing clonal cell lines were transferred to 125 ml Erlenmeyer flasks with the ProCHO 5 culture medium supplemented with 4 mM glutamine, 4 mM alanyl-glutamine and 8 μM MTX and after 5 days in suspension culture, the best producing clone was determined by measuring the product titer and cell concentration.

## Small-scale protein purification

Harvested cell suspension, up to 150 ml, was clarified by 5 minutes centrifugations at 2000 g to remove cells and subsequent 15 minutes centrifugation of supernatant at 20000 g. The clarified supernatant was concentrated by ultrafiltration to approximately 70 ml using one 200 cm$^2$ 10 kDa MWCO PES tangential flow filtration cassette (Sartorius, Germany) and the ultrafiltration reservoir (Sartorius). Concentrated cell supernatant was diafiltered with 500 ml of the PBS, then adjusted to 300 mM NaCl and used for chromatographic purification.

Chromatography was performed at room temperature by the Akta Explorer system (Cytiva, USA) and the Chelating Sepharose Fast Flow 1 ml disposable HiTrap column (Cytiva), charged by Ni$^{2+}$ ions or 1 ml Tricorn 5/10 column (Cytiva) packed by Ni-NTA resin (Thermo Fisher Scientific). Columns were equilibrated by the 50 mM Na-PO$_4$, pH = 8.0; 500 mM NaCl solution, sample application was performed at 1 ml/min, columns were washed by the equilibration solution until stable baseline (approximately 10 column volumes) at 2 ml/min flow velocity and subjected to stepwise elution at 1 ml/min, utilizing equilibration solution with 50–250 mM imidazole-HCl, pH 8.0 added. Columns were stripped by the 50 mM EDTA-Na, pH 8.0 solution.

## Preparative protein purification

One liter of harvested cell suspension was clarified by 5 minutes centrifugation at 2000 g to remove cells and subsequent 15 minutes centrifugation of supernatant at 20000 g. The clarified supernatant was concentrated approximately fivefold by ultrafiltration using the QuixStand apparatus with the peristaltic pump (Cytiva), ultrafiltration cassettes holder (Sartorius) and three 5 kDa MWCO cassettes (PES, Millipore, 0.1 m$^2$). Concentrated cell supernatant was buffer exchanged by diafiltration by 10 mM imidazole-HCl, pH 8.0 solution until constant filtrate conductivity was achieved, approximately 15 diafiltration volumes. Desalted culture supernatant was supplemented by sodium phosphate, pH 8.0 and sodium chloride solutions, final concentrations– 50 mM Na-PO$_4$ and 500 mM NaCl.

Chromatographic protein purification was performed at room temperature by the Akta Explorer system (Cytiva) and the XK 16/20 column (Cytiva), packed with 7 ml Ni-NTA Agarose resin (R90101, Thermo Fischer Scientific). Column was equilibrated by 50 mM sodium phosphate, pH 8.0; 500 mM NaCl and 10 mM imidazole-HCl, pH 8.0 solution. Desalted culture supernatant was applied to the column at 3.5 ml/min (2 minutes contact time). The column was washed by equilibration solution until a stable baseline at 7 ml/min flow (approximately 10 column volumes), then by the equilibration solution with 50 mM imidazole-HCl at the same flow velocity for 10 column volumes. The target protein was eluted by the equilibration solution with 300 mM imidazole-HCl at 3.5 ml/min; the eluate was collected as a single

fraction. The flow was paused for 5 minutes until the first 7 ml of eluate were collected for minimization of total elution volume.

Eluted purified RBD protein solutions were desalted by the 200 cm$^2$ disposable tangential flow filtration cassette, Hydrosart, 10 kDa MWCO (Sartorius), connected to the diafiltration reservoir (Sartorius) and the peristaltic pump (MasterFlex, USA). The diafiltration solution was 10 mM sodium phosphate, pH 7.4; diafiltration was carried out until constant conductivity of the filtrate, approximately 20 diafiltration volumes. Desalted RBD solutions were finally concentrated to the 3–8 mg/ml by ultrafiltration centrifugal concentrators, 10 kDa MWCO PES membranes (Sartorius); supplemented by NaCl, 140 mM final concentration, aliquoted and stored frozen at -70˚C.

## Sodium Dodecyl Sulphate-polyacrylamide gel electrophoresis (SDS-PAGE)

SDS-PAGE was performed with the 12.5% acrylamide in the separating gel, in reducing conditions, if not stated otherwise, with the PageRuler prestained marker, 5 μl/lane (ThermoFisher Scientific). Gels were stained by the colloidal Coomassie blue according to [27], scanned by the conventional flatbed scanner in the transparent mode as 16-bit grayscale images and analyzed by the TotalLab TL120 gel densitometry software (Nonlinear Dynamics, UK).

## Western blot

SDS-PAGE was performed as described above, protein transfer, blocking, hybridization and color development were done according to [28] using nitrocellulose transfer membrane (GVS Group, Bologna, Italy) and Towbin buffer with methanol. Primary anti-c-myc antibody (SCI store, Moscow, Russia #PSM103-100) was used at the 1:2000 dilution, anti-mouse-HRP conjugate (Abcam, Cambridge, UK, ab6789) was used at 1:2000 dilution; membrane was developed by the DAB-metal substrate and scanned by the flatbed scanner in the reflection mode.

## Analytical size-exclusion chromatography

Multimeric forms of the RBD were quantified by size exclusion chromatography, utilizing Waters Alliance 2695 chromatography apparatus (Waters, USA) with the PDA detector and the Superdex 200 GL 10/30 column (Cytiva). Mobile phase was 1xPBS + 0.02% sodium azide, flow 0.5 ml/min, injection volume– 30 μl. All samples were analyzed in duplicates. Molecular size calibrators used were chimeric IgG1 infliximab (Roche, Bazel, Switzerland), bovine serum albumin (Merck), follicle stimulating hormone (produced in-house according to [24]) and RNAse A (Merck). Absorbance was detected at 280 nm. Chromatography traces were analyzed by Waters Millenium software (Waters), the molecular mass of unknowns was calculated by linear regression of the elution time–log(MW) chart.

## Peptide:N-glycosidase F treatment

Ten micrograms of purified RBD proteins were treated by 500 units of PNGase F (New England Biolabs, Ipswich, USA #P07045) according to manufacturer's instruction, denaturing protocol. Five micrograms of deglycosylated proteins were analyzed by SDS-PAGE, the rest was used for mass-spectrometry analysis.

## Mass spectrometry

Tryptic peptides were obtained from RBDv1 and RBDv2 bands, excised from the SDS-PAGE gel and sliced into pieces. The gel pieces were de-stained with 2 washes by 200 μl 50% acetonitrile (ACN), 50 mM ammonium bicarbonate solution (ABC). Destained gel pieces were dried

*in vacuo* and rehydrated by the 12.5 μg/ml trypsin (Promega, USA), 50 mM ABC, 5 mM CaCl$_2$ solution. Proteolytic digestion was carried out for 16 h at 37°C. Peptides were extracted from the gel with 25 mM ABC, following by the 80% ACN. Extracts were vacuum-dried and redissolved in the 0.5% trifluoroacetic acid (TFA), 3% ACN. Prepared solutions were mixed at 3:1 ratio with 20% α-cyano-4-hydroxycinnamic acid (Merck) solution in 20% ACN, 0.5% TFA on the target plate.

Solutions of intact and deglycosylated proteins were passed through the ZipTip C18 microcolumns (Millipore), washed and eluted according to manufacturer protocol. One and a half μl of protein solutions were mixed on the target plate with 0.5 μl of the 20% 2,5-dihydroxybenzoic acid (Merck) solution in 20% ACN, 0.5% TFA. Mass spectra were obtained by the MALDI-TOF mass spectrometer Ultraflextreme Bruker (Germany) with the UV-laser (Nd), linear mode, positive ions. Spectra were obtained in the 500–5000 Da range for tryptic peptides mixtures, 5000–50000 Da range for intact proteins.

Mass lists for each sample were calculated by the Bruker Daltonics flexAnalysis software (Germany), peptides identification was performed by the GPMAW 4.0 software (Lighthouse data, Denmark) and by the Mascot server (Matrix Science, Boston, USA). Glycopeptides mass assignment was performed by the GlycoMod online software tool [29].

### Enzyme-linked immunosorbent assay (ELISA)

Sandwich ELISA with anti- S protein antibodies was performed using a prototype of the SARS-CoV-2 antigen detection kit (Xema Co., Ltd., Moscow, Russia, a generous gift of Dr. Yuri Lebedin). Pre-COVID-19 normal human plasma sample (Renam, Moscow, Russia) was used for preparation of the SARS-CoV-2 negative serum sample. Control pooled serum samples obtained from patients with the PCR-confirmed SARS-CoV-2 infection and distributed by the Xema ltd were tested as positive sample.

Antibody capture ELISA with human serum samples was performed according to [28] at the 100 ng per well antigens load. Antigens were applied on ELISA 96-well plates (Corning, USA) overnight at + 4oC, in PBS, 100 μl/well. Plates were washed by PBS– 0.02% Tween (PBST) thrice and blocked by 250 μl/well of the 3% BSA in PBS solution, washed by PBST and used immediately. Test sera were inactivated by heating at 56°C for 30 min, diluted by the 1% BSA-PBS, applied as serial dilutions in the 1:20–1:82 000 range, and incubated for 1 h at 37°C. Wells were washed three times by PBST, secondary anti-human IgG antibody-HRP conjugate (Xema Co., Ltd., cat. T271X@1702) was used at the 1:20 000 dilution, the incubation time was 1 h at +37°C. Wells were washed five times by the PBST. The color was developed for exactly 10 minutes at room temperature (+25±2 °C), utilizing the ready-to-use TMB solution (Xema Co., Ltd.), 200 μl/well. The reaction was stopped by adding 100 μl of 5% orthophosphoric acid. Optical densities were measured by the Multiskan EX plate reader (Thermo Fischer Scientific) at 450 nm. All samples were tested in duplicates or triplicates.

### Statistical analysis

The t-test was performed using the GraphPad QuickCalcs Web site: https://www.graphpad.com/quickcalcs/ttest1.cfm (accessed November 2020).

## Results

### SARS-CoV-2—RBDv1

The native N-terminal signal peptide of SARS-CoV-2 S protein (amino acid sequence MFVFLVLLPLVSSQ) was fused to the RBD sequence (319–541, according to [NCBI

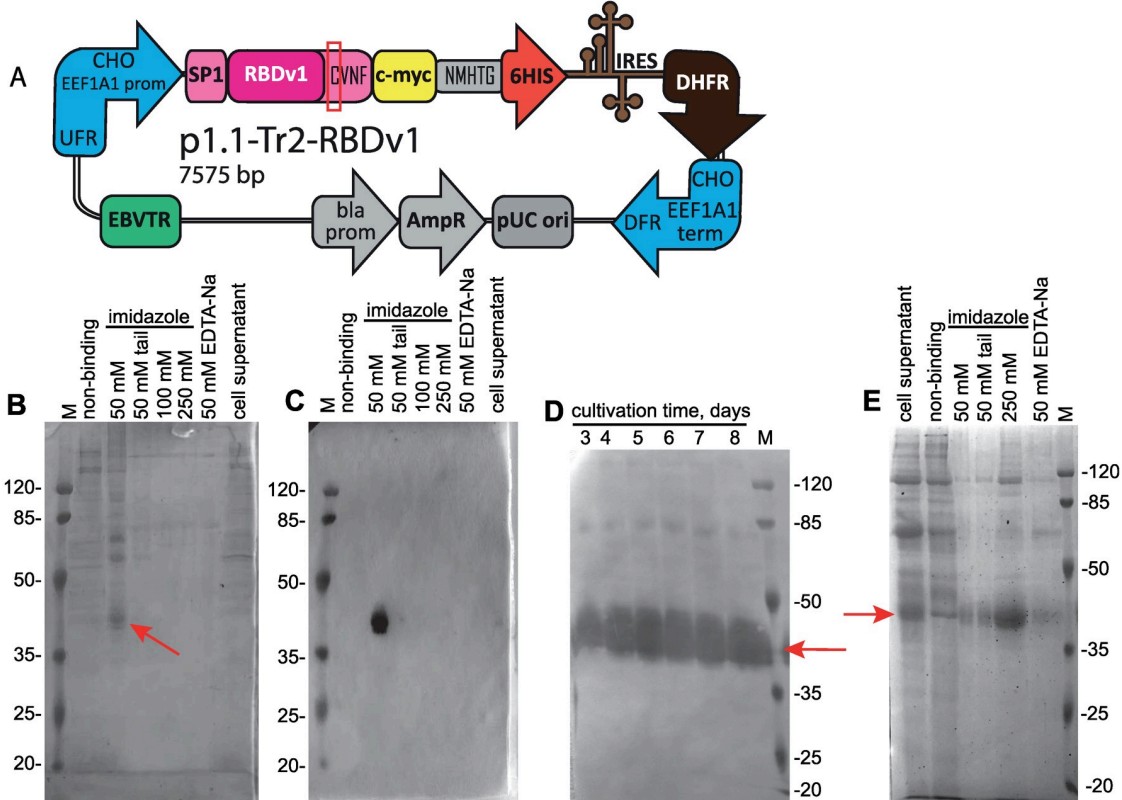

**Fig 1. Map of the p1.1-Tr2-RBDv1 plasmid, purification of the RBDv1 protein its extended batch cultivation.** A—Scheme of p1.1-Tr2-RBDv1 expression plasmid. CHO EEF1A1 UFR–upstream flanking area of the *EEF1A1* gene (*EEF1A1* gene promoter, flanked with 5' untranslated region), DFR–downstream flanking area (*EEF1A1* gene terminator and polyadenylation signal, flanked with 3' untranslated region); IRES–encephalomyocarditis virus internal ribosome entry site; DHFR–ORF of the dihydrofolate reductase gene of *Mus musculus*, pUC ori–replication origin; AmpR and bla prom–ampicillin resistance gene and the corresponding promoter; EBVTR–fragment of the long terminal repeat from the Epstein-Barr virus; SP1 –signal peptide, RBDv1 –RBD ORF, amino acids from natural Spike protein sequence (SP1 and CVNF) are shown in pink color; unpaired cysteine is surrounded by a red rectangle; c-myc and 6HIS–C-terminal fusion tags, NMHTG—linker peptide sequence. B–SDS-PAGE analysis of the RBDv1 purification by the IDA-Ni chromatography resin. Protein was obtained from the cell culture at 2 μM MTX selection pressure, cell supernatant and non-binding proteins fractions applied without ultrafiltration, 10 μl/lane, column elution fractions applied as 10 μl from 5 ml total, concentration factor 30x. Position of the target protein band is depicted by arrow. Molecular masses are given in kDa. C–Western blot analysis of the RBDv1 purification by the IDA-Ni chromatography resin. Anti c-myc primary antibodies, lanes load same to panel B. D—Western blot analysis of the extended batch cultivation of the RBDv1 2 μM MTX cell population, 2 μl of cell supernatant/lane. E—SDS-PAGE analysis of the RBDv1 purification by the Ni-NTA chromatography resin. Lane loads same to panel B.

Proteins ID: YP_009724390.1]) and joined with a C-terminal c-myc epitope (EQKLISEEDL), short linker sequence, and hexahistidine tag. N-terminal part of the RBDv1 gene was constructed according to [13], utilizing the optimized codon usage gene structure. C-terminal tags were not optimized for codon usage frequencies. The resulting synthetic gene was cloned into the p1.1-Tr2 vector plasmid (S1 File), a shortened derivative of the p1.1 plasmid [21], and used for transfection of DHFR-deficient CHO DG44 cells. The resulting expression plasmid p1.1-Tr2-RBDv1 [GenBank: MW187858] (S2 File) is shown on Fig 1A. The stably transfected cell population was obtained by selection in the presence of 200 nM of DHFR inhibitor methotrexate, RBD titer 0.33 mg/L was detected for 3-days culture (S1 Fig in S1 Raw file). One-step target gene amplification was performed by increasing the MTX concentration tenfold and maintaining the cell culture for 17 days until cell viability restored to

more than 85%; the resulting polyclonal cell population could secrete up to 3.0 mg/L RBD in the 3-days culture. The target protein was purified by a single immobilized metal affinity chromatography (IMAC) step, utilizing the IDA-based resin Chelating Sepharose Fast Flow (Cytiva), Ni2+ ions, and step elution by increasing imidazole concentrations (Fig 1B and 1C). The resulting protein production method was found to be sub-optimal due to unexpectedly low secretion rate, signs of cellular toxicity of the target gene– 33 h cell duplication time, maximal cell density in shake flask of 2.3 ×10 exp6 cells/ml (S2 Fig in S1 Raw file), and unacceptable level of contaminant proteins co-eluting with the RBDv1. At the same time, the RBDv1 protein was stable in the culture medium during the extended batch cultivation of cells for at least 7 days (Fig 1D), making the long-term feed batch cultivations a viable option for its production in large quantities.

We proposed that target protein secretion rate and its purity after one-step purification could be significantly improved by a simultaneous shift of the RBD domain boundaries, exchange of the SARS-CoV-2 S protein native signal peptide to the signal peptide of more abundantly expressed protein, two-step genome amplification and switch from IDA-based resin to the NTA-based one (Fig 1E).

## SARS-CoV-2—RBDv2

Human tissue plasminogen activator signal peptide (hTPA SP, amino acid sequence MDAMKRGLCCVLLLCGAVFVSAS) is commonly used for heterologous protein expression in mammalian cells. It was successfully used for the expression of SARS-CoV S protein in the form of DNA vaccine [30] and envelope viral protein gp120 [31]. In the case of MERS-CoV S protein RBD–Fc fusion protein, various heterologous signal peptides modulate target protein secretion rate by the factor of two [14].

Corrected boundaries of the SARS-CoV-2 RBD were determined according to the cryo-EM data [PDB ID: 6VXX] [32] obtained for the trimeric SARS-CoV-2 S protein ectodomain. Initially used 319–541 coordinates, described in the [13] include one unpaired Cys residue originated from the N-terminal part of the next domain SD1 (structural domain 1), so we excluded Lys319 from the N-terminus of the mature RBD protein, aiming at the maximization of signal peptide processing, and removed C-terminal aminoacids $C_{538}VNF_{541}$, which form the structure of the SD1 domain. Both linker areas surrounding the folded RBD domain core remain present in the RBDv2 protein (320–537, according to the [Genbank: YP_009724390.1]). Additionally, we redesigned C-terminal tags by introducing the Pro residue immediately upstream of the c-myc tag, adding the short linker sequence SAGG between the c-myc tag and polyhistidine tag, and extending the polyhistidine tag up to 10 residues. We expected this structure to expose the c-myc tag properly on the protein globule's surface and move the decahistidine tag away from possible masking negatively charged protein surface areas.

We constructed an expression vector pTM [GenBank: MW187855] (S3 File), where consensus Kozak sequence, hTPA SP and c-myc and 10-histidine tags are coded in the polylinker. RBD coding fragment was cloned in-frame, resulting pTM-RBDv2 expression plasmid [GenBank: MW187856] (S4 File) is shown on Fig 2A.

CHO DG44 cells were transfected by the pTM-RBDv2 plasmid; the stably transfected cell population was established at the 200 nM MTX. Target protein titer was similar to the previous plasmid design– 0.9 mg/L for 3-days culture, but after one step of the MTX-driven genome amplification, it increased eleven-fold to 9.7 mg/L at 2 μM MTX (Fig 2B) and then increased by a factor of 2.5 after second amplification step at 8 μM MTX, resulting titer was 24.6 mg/L for 3-days culture (Fig 3A). A steady increase of the target protein titer was detected for the extended batch cultivation of polyclonal cell population obtained at 8 μM MTX, peaking at 50

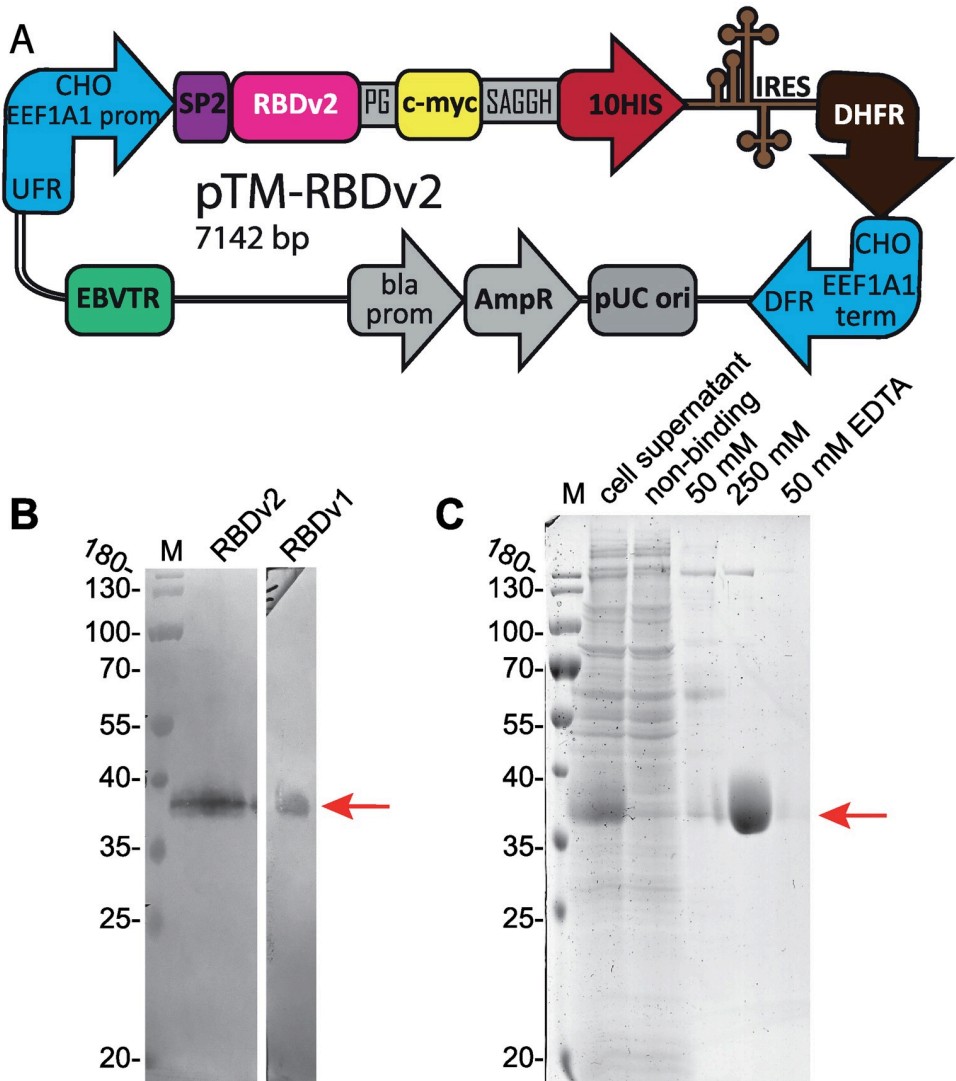

**Fig 2. Map of the pTM-RBDv2 plasmid and purification of the RBDv2.** A—Scheme of pTM- RBDv2 expression plasmid. CHO EEF1A1 UFR–upstream flanking area of the *EEF1A1* gene (*EEF1A1* gene promoter, flanked with 5' untranslated region), DFR–downstream flanking area (*EEF1A1* gene terminator and polyadenylation signal, flanked with 3' untranslated region); IRES–encephalomyocarditis virus internal ribosome entry site; DHFR–ORF of the dihydrofolate reductase gene of *M. musculus*, pUC ori–replication origin; AmpR and bla prom–ampicillin resistance gene and the corresponding promoter; EBVTR–fragment of the long terminal repeat from the Epstein-Barr virus; SP2 –hTPA signal peptide, RBDv2 –RBD ORF; c-myc and 10HIS–C-terminal fusion tags, PG and SAGGH—linker peptide sequence. B–Western blot analysis of the RBDv2 in the culture medium, 16 μl of cell supernatant were obtained from the RBDv2 2 μM MTX culture. Control RBDv1 protein was used in purified form, 100 ng/lane. C–SDS-PAGE analysis of the RBDv2 purification by the Ni-NTA chromatography resin. 50 mM, 250 mM–eluates, obtained at 50 mM and 250 mM imidazole concentrations.

mg/L at 8 days of cultivation in the 2 L shake flask (Fig 3B and 3C). A similar ratio of product titer increase after multi-step MTX-driven genome amplification was described for the MERS-CoV RBD– 40-fold increase after 9 steps of consecutive increments of MTX concentration, overall amplification period length was 60 days [14]. Vector plasmid pTM, used in this study, allowed a much more rapid amplification course–a 27-fold titer increase in two steps, 33 days total. This RBD variant was less toxic for cells, as the 2 μM MTX cell culture attained a

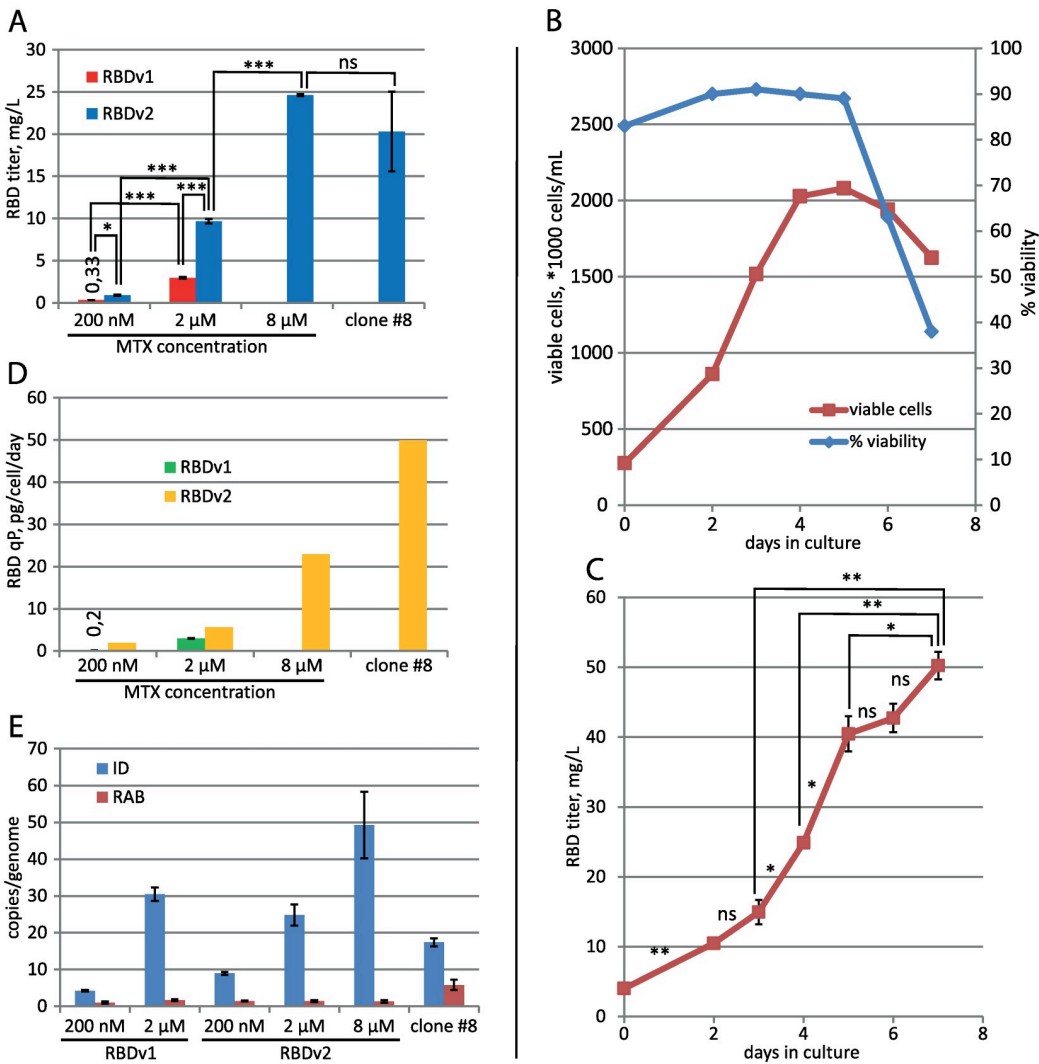

**Fig 3. Productivity dynamics of cell populations and clonal lines, secreting RBD proteins.** A–Concentration of target proteins in cell supernatant by ELISA. 3-day cultures, all seeded at 3–4 ×10exp5 cells/ml. Data are shown as mean ± SD, n = 2. *—p <0.05, t-test; **—<0.01, ***—<0.001, ns–p > = 0.05. B–Cell culture growth curve for the preparative cultivation in 2 L Erlenmeyer flask, RBDv2 8 μM MTX cell population. C–RBDv2 titer dynamics by ELISA, preparative culture in 2 L Erlenmeyer flask, RBDv2 8 μM MTX cell population. Data are shown as mean ± SD, n = 2, t-test for pairs of cell supernatants, taken on two subsequent days. D–Specific productivities of cell populations. RBD secretion level in the stably transformed cell populations was determined by ELISA. E–qPCR analysis of the expression cassette copy numbers per haploid genome for RT-ID-F, RT-ID-R primer pair (ID); Rab1 copy numbers for the control sequence (RAB), supposedly unique to the CHO genome, RT-Rab1-F, RT-Rab1-R primer pair, n = 4. All values for ID primer pair are significantly different from each other, t-test.

higher cell density of 4.4 ×10exp6 cells/ ml in shake flask with only 25 h cell duplication time. The clonal cell line, secreting the RBDv2 protein, was obtained from the 8 μM MTX–resistant polyclonal cell population by limiting dilution. Clone #8, one of the 12 clones screened, possessed a very high specific productivity, 50 pg/cell/day (Fig 3D), and an acceptable cell doubling time of 48 h. This clonal cell line can be used to produce large quantities of RBDv2 with anticipated high uniformity in the pattern of post-translational modifications.

The quantitative PCR analysis of all RBD-secreting cell populations showed that increased productivity of populations adapted to higher MTX concentrations corresponds to higher

copy numbers of the target gene (Fig 3E). At the same time, higher cell productivity for RBDv2 was achieved not due to an increase in gene copy number compared with RBDv1.

Cell culture medium Pro CHO5 (Lonza), utilized in this study, contains unknown components, blocking His-tagged RBD protein's interaction with the Ni-NTA chromatography resin. Clarified conditioned medium, used for protein purification, was concentrated approximately tenfold by tangential flow ultrafiltration on the 5 kDa MWCO cassettes and completely desalted by diafiltration, 20 diafiltration volumes of the 10 mM imidazole-HCl, pH 8.0 solution. RBDv1 and RBDv2 proteins were purified by IMAC utilizing Ni-NTA Agarose (Thermo Fischer Scientific, USA) in the same conditions. Desalted conditioned medium was applied onto the column in the presence of 10 mM imidazole; the column was washed by the solution containing 50 mM imidazole, it removed the majority of admixtures and only minor quantities of the target proteins. In the case of Ni-IDA ligand, interaction with the 6xHis tag on the RBDv1 protein was insufficient to retain the target protein on the column in 50 mM imidazole, resulting in premature elution of RBDv1 along with contaminating cell proteins. In the case of Ni-NTA ligand, both 6xHis and 10xHis tags were suitable for efficient removal of proteinaceous admixtures. This observation is in line with the general properties of immobilized metal affinity chromatography resins. [33]. Elution was performed by the 250–300 mM imidazole solution; further column strip by the 50 mM EDTA-Na solution revealed no detectable target protein RBDv2 in the eluate (Fig 2C). Purified proteins were desalted by another round of ultrafiltration/diafiltration on the centrifugal concentrators with 5 kDa MWCO membranes; diafiltration solution was PBS; final concentration 3–7 mg/ml. Purified proteins were flash-frozen in liquid nitrogen and stored frozen in aliquots. Overall protein yield for RBDv2 was 64%, 32 mg of purified RBDv2 were obtained from 1 L shake flask culture.

## Analysis of purified proteins

The apparent molecular weight of the RBDv1 estimated by the SDS-PAGE mobility was 35.3 kDa for the intact protein and 26.1 kDa for the deglycosylated RBDv1; its theoretical molecular weight is 27647 Da. For the RBDv2 the apparent molecular weight was 35.7 kDa for the intact protein and 28.5 kDa for the deglycosylated protein; its theoretical molecular weight is 27459 Da (Fig 4A). Both protein variants supposedly possess two distinct forms of intramolecular disulfide bonds sets. This variation is visible as pairs of closely adjacent protein bands in non-reducing conditions and the complete absence of such band pattern in reducing conditions for RBDv2 protein variant. Detailed analysis of this phenomenon will require the complex MS/MS analysis of Cys-linked tryptic peptides of the RBD variants and is beyond the limits of the current study.

Previously it was reported that SARS-CoV-2 RBD 319–541, expressed transiently in HEK-293 cells, tends to form a covalent dimer, around 30% from the total, visible as the 60 kDa band on the denaturing gel in non-reducing conditions [19]. We confirmed this observation; in the case of stably transfected CHO cells, covalent dimerization was also 31%, according to gel densitometry data. At the same time, it should be noted that the RBDv2 protein, redesigned explicitly for mitigation of this unwanted dimerization and containing an even number of Cys residues, still forms 6% of the covalent dimer.

Purified RBDv2 was tested by size exclusion chromatography. The apparent molecular weight of the major monomer form was 32.4 kDa (S3 Fig in S1 Raw file); the apparent molecular masses of the admixtures peaks corresponded well to RBD dimer, tetramer, and two high molecular mass oligomers accounting for 6% of all peak areas (Fig 4B).

Mass-spectrometry analysis of RBDv1 and RBDv2 revealed that both proteins' molecular masses diminished after PNGase F treatment by approximately 3200–3500 Da (S4 Fig in

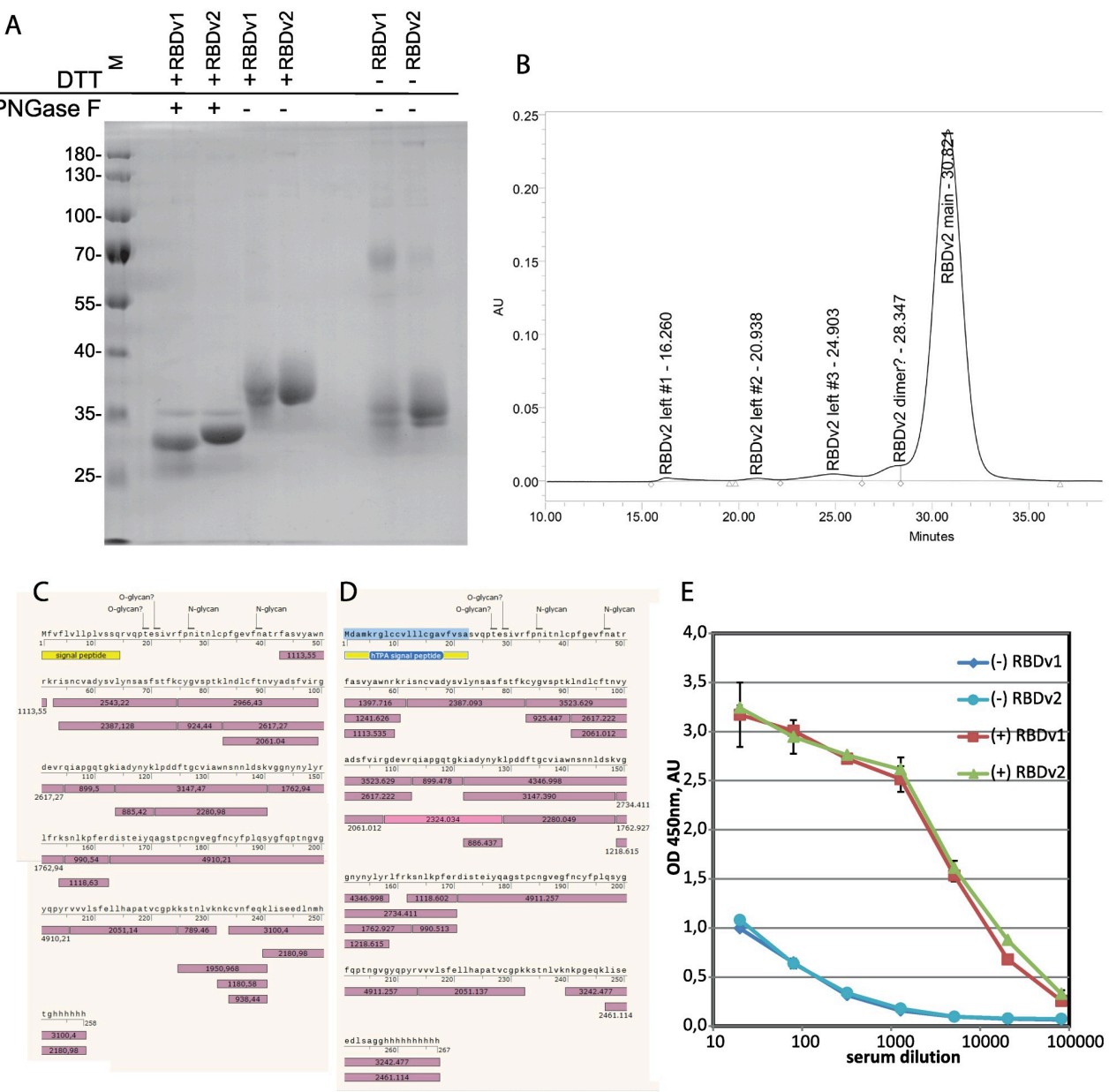

**Fig 4. Analysis of purified RBDv1 and RBDv2 proteins.** A–SDS-PAGE for PNGase F–treated RBDv1 and RBDv2 proteins and non-reduced proteins. B–Size exclusion chromatography trace for the RBDv2 protein. C, D–protein sequence coverage by tryptic peptides, MALDI-TOF analysis. Glycosylated peptides found are not pictured, signal peptides are yellow, detected tryptic peptides–violet, experimentally obtained masses, [M+H]+, are stated in the boxes. E–Immunoreactivity of RBDv1 and RBDv2 by ELISA with pooled serum samples from PCR-positive patients–(+), pre-COVID-19 pooled sera (-). All sera samples were analyzed in duplicates, data are mean ± SD.

S1 Raw file); these mass shifts correspond to the removal of two N-linked oligosaccharide groups. The molecular masses of deglycosylated RBDv1 and RBDv2 were 28781 Da and 28123 Da, the theoretical molecular weights of polypeptides—27263 and 27548 Da. We suppose that both protein variants have O-linked glycans. This hypothesis is supported by the complete absence of the signal from the N-terminal tryptic peptide (S)V$_{339}$QPTESIVR$_{347}$ for both protein variants. This peptide was found to contain O-glycans in the systemic LC-MS/MS scan of the full-length ectodomain of the S protein [10]. In silico search of possible glycopeptides with

O-linked glycans for RBDv1 and RBDv2 tryptic peptides mixtures revealed two pairs of candidate structures—(Hex)1 (HexNAc)1 and (Hex)1 (HexNAc)1 (NeuAc)1, presented in both mass spectra (S5 Table in S1 Raw file). These structures were directly detected in the $V_{339}$QPTESIVR$_{347}$ peptide by the LC-MS/MS analysis of the S protein.

Both RBD protein variants were fully covered by tryptic peptides obtained by the in-gel digestion (Fig 4B and 4C), except for N-terminal peptides, found only in the form of glycopeptides and subsequent $F_{348}$PNITNLCPFGEVFNATR$_{365}$ peptide, containing two N-glycosylation sites in positions N350 and N362 (S5 and S6 Figs in S1 Raw file). This long peptide was completely absent in both spectra of deglycosylated proteins (S1–S4 Tables in S1 Raw file). A more detailed analysis of this area of the RBD protein may be of some interest for the S protein structure-function investigation but is out of scope for the present study.

Purified RBD variants were used as antigens for microplates coating and subsequent direct ELISA with pooled sera obtained from patients with the RT-PCR-confirmed COVID-19 diagnosis, and serum sample obtained from healthy donors before December 2019 (Fig 4E). Both RBD variants perform equally–all serum samples produce highly similar OD readings for all dilutions tested with both antigens.

## Discussion

We describe a method of generating stably transfected CHO cell lines, secreting large quantities of monomeric SARS-CoV-2 RBD, suitable for serological assays. Serological assays for detecting seroconversion upon SARS-CoV-2 infection are mostly based on two viral antigens–N and S proteins, or fragments of the S protein, including the RBD. In some cases, the sensitivity of clinically approved N-based assays was challenged by direct re-testing of N-negative serum samples by the RBD-based assays [34]. Other studies question N-based ELISA tests' specificity, demonstrating a significant level of false-positive results for the full-length SARS-CoV-2 N-antigen [35]. The most accurate results can be presumably achieved by testing of the serum samples with both SARS-CoV-2 antigens, as was done, for example, in the South-East England population study [36]; the same conclusion was made in the microarray study of a limited number of patients serum samples [37].

It is unclear yet, which part of the S protein is the optimal antigen for serological assays; microarray analysis revealed that S2 fragment generates more false-positive results than S1 or RBD antigen variants [37] in the case of IgG detection; at the same time, the RBD protein generated much lower signals on COVID-19 patients serum samples then S1 or S1+S2 antigens. In another microarray study it was found that IgG response toward the RBD domain in the convalescent plasma samples correlates well with the response toward full-length soluble S protein [38]. In the conventional ELISA test format, RBD demonstrated nearly 100% specificity and sensitivity on a limited number of SARS-CoV-2 patients and control serum samples [2]. As of 26.10.20, at least 104 various immunoassays for SARS-CoV-2 antibodies were authorized for *in vitro* diagnostic use in the EU [39], many of them use RBD as the antigen. A simple ELISA screening test with the 96-well microplate will consume around 10 µg of the RBD antigen for 40 test samples, so even one million tests will require 250 mg of the purified RBD protein, making the antigen supply a critical step in the production of such tests. Method of the generation of highly productive stably transfected CHO cell line, secreting the RBD protein, can be useful for manufacturers of diagnostic tests, who need a source of antigen with constant properties.

Although the RBD fragment of the S-protein from SARS-CoV-2 is not the most popular antigen variant in the current efforts of anti-SARS-CoV-2 vaccine development [40], it can be considered as the viable candidate for a simple subunit vaccine. It demonstrated the significant

protective immune response development in rodents, without signs of antibody-dependent enhancement effect [41], and some RBD-based protein subunit vaccines have advanced to Phase II clinical trials. Cultured CHO cells are the reliable source of RBD protein for this kind of vaccine; at the productivity level achieved in our study, only 30 $m^3$ of cell culture supernatant will provide enough antigen material for 100 mln of typical 10 mg/vial vaccine doses.

The described method for obtaining the 320–537 RBD fragment of the SARS-CoV-2 S-protein can be immediately employed to produce a highly consistent RBD reagent for serological assays. The same cell line creation strategy will be applied to obtain RBD 320–537 protein without C-terminal tag, potentially suitable for subunit vaccine studies. This variant of the RBD protein will be characterized in much more detail, including direct analysis of N-linked and O-linked glycans, peptide mapping with S-S bridges positions quantification and surface plasmon resonance studies of interaction with the ACE2 receptor and known neutralizing antibodies. Highly productive RBD-secreting CHO cell lines are already mentioned in the SARS-CoV-2 RBD dimer vaccine candidate studies but without any details on the cell line generation methodology [42].

Other S-protein domains and multi-domain subunits, suitable for serological assays or vaccine studies, can be obtained in stably transfected CHO cells using the vector plasmid pTM described in this study. Although we found no visible changes in the properties of the RBD 319–541 and RBD 320–537 variants used as ELISA antigens, there is a sharp difference in the antigenic properties of the RBD monomer and dimer, described in [42]–dimeric beta-CoV RBD produced much higher titers of neutralizing antibodies due to unknown reasons. Thus, SARS-CoV-2 RBD preparations with a high content of the covalent dimer admixture, i.e., 319–541 variant, can produce incorrectly positive results in the subunit vaccine studies; this variant should be replaced by the shorter 320–537 RBD variant, containing much lower content of the covalent dimer admixture.

## Supporting information

**S1 File. p1.1-Tr2-eGFP [Genbank MW187857].** p1.1-Tr2-eGFP plasmid sequence and feature map.
(GB)

**S2 File. p1.1-Tr2-RBDv1 [Genbank MW187858].** p1.1-Tr2-RBDv1plasmid sequence and feature map.
(GB)

**S3 File. pTM [Genbank MW187855].** pTM plasmid sequence and feature map.
(GB)

**S4 File. pTM-RBDv2 [Genbank MW187856].** pTM-RBDv2 plasmid sequence and feature map.
(GB)

**S1 Raw images.**
(DOCX)

**S1 Raw file.**
(DOCX)

## Acknowledgments

We thank Mr. Arthur Isaev (Genetico, Moscow, Russia) and Dr. Alexander Ivanov (Institute of Molecular biology Russian Academy of Sciences, Moscow, Russia) for valuable comments

and early access to the SARS-CoV-2 S protein sequence data, Dr. Yuri Lebedin, Eugenia Kostrikina and Xema Co., Ltd., for providing anti-RBD mAbs conjugates and control sera samples.

The measurements were carried out in the Shared-Access Equipment Centre "Industrial Biotechnology" the Research Center of Biotechnology of the Russian Academy of Sciences. DNA sequencing was carried out in the inter-institutional Center for collective use "GENOME" IMB RAS, organized with the support of the Russian Foundation of Basic Research. The authors would like to acknowledge all the doctors who diagnose and treat patients during the COVID-19 pandemic.

## Author Contributions

**Conceptualization:** Nadezhda A. Orlova, Ivan I. Vorobiev.

**Data curation:** Nadezhda A. Orlova.

**Formal analysis:** Maria V. Sinegubova, Nadezhda A. Orlova, Ivan I. Vorobiev.

**Investigation:** Maria V. Sinegubova, Nadezhda A. Orlova, Sergey V. Kovnir, Lutsia K. Dayanova, Ivan I. Vorobiev.

**Methodology:** Maria V. Sinegubova, Nadezhda A. Orlova, Sergey V. Kovnir, Ivan I. Vorobiev.

**Project administration:** Nadezhda A. Orlova.

**Supervision:** Ivan I. Vorobiev.

**Visualization:** Nadezhda A. Orlova.

**Writing – original draft:** Maria V. Sinegubova, Nadezhda A. Orlova, Ivan I. Vorobiev.

**Writing – review & editing:** Nadezhda A. Orlova, Ivan I. Vorobiev.

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
