## [Decision Letter · Decision Letter 0]

11 Dec 2020

PONE-D-20-36259

High-level expression of the monomeric SARS-CoV-2 S protein RBD 320-537 in stably transfected CHO cells by the *EEF1A1*-based plasmid vector.

PLOS ONE

Dear Dr. Orlova,

Thank you for submitting your manuscript to PLOS ONE. After careful consideration, we feel that it has merit but does not fully meet PLOS ONE’s publication criteria as it currently stands. Therefore, we invite you to submit a revised version of the manuscript that addresses the points raised during the review process by both the reviewers.

We look forward to receiving your revised manuscript.

Kind regards,

Paulo Lee Ho, Ph.D.

Academic Editor

PLOS ONE

Journal Requirements:

3.We note that you have a patent relating to material pertinent to this article. Please provide an amended statement of Competing Interests to declare this patent (with details including name and number), along with any other relevant declarations relating to employment, consultancy, patents, products in development or modified products etc. Please confirm that this does not alter your adherence to all PLOS ONE policies on sharing data and materials, as detailed online in our guide for authors http://journals.plos.org/plosone/s/competing-interests by including the following statement: "This does not alter our adherence to  PLOS ONE policies on sharing data and materials.” If there are restrictions on sharing of data and/or materials, please state these. Please note that we cannot proceed with consideration of your article until this information has been declared.

Reviewers' comments:

Reviewer's Responses to Questions

**Comments to the Author**

1. Is the manuscript technically sound, and do the data support the conclusions?

Reviewer #1: Yes

Reviewer #2: Yes

2. Has the statistical analysis been performed appropriately and rigorously? 

Reviewer #1: Yes

Reviewer #2: Yes

3. Have the authors made all data underlying the findings in their manuscript fully available?

Reviewer #1: Yes

Reviewer #2: Yes

4. Is the manuscript presented in an intelligible fashion and written in standard English?

Reviewer #1: Yes

Reviewer #2: Yes

5. Review Comments to the Author

Reviewer #1: The authors developed 2 versions of their unique EEF1A1-based vector for the expression of RBD protein of SARS-CoV-2 differing in relation to signal peptide and sequence that was optimized for version 2. Each vector was transfected in CHO DG44 cells and high level expression and stable cell line was observed in RBDv2 variant. The purified protein was obtained by simple purification strategy with high yield and it was well characterized.

The EEF1A1-based vector developed by the group was applied for the production of other recombinant proteins and the authors obtained high expression on their strategies, showing that the vector is a powerful tool to attend an emergency demand to obtain new recombinant proteins not available.

Up to now the expression of glycosylated RBD mentioned in the papers considering mammalian cells was in HEK293 cells by transient transfection. The development of high-producer and stable cell lines in CHO cells will be very important for the application of RBD in serological assays at large scale. Obtainment of cell line that produces high level expression of glycosylated RBD protein will be able to contribute for the development of serological assay that would attend more people in the world. The authors performed ELISA using both types of RBD and they tested for negative and positive sera for anti-SARS-CoV-2. The result was good and similar for both versions of RBD. However, there is no mention of perspectives and next steps for the present work by the authors.

I suggest the authors to include a conclusion paragraph at the end of “Discussion” section or add “Conclusion section” to sum up the importance of the results obtained in the manuscript.

Other considerations:

Line 52-53: Explain the last sentence better. RBD produced was tested in ELISA and the authors mention “the described technique is suitable for serological tests and similar applications.” What is the meaning of “similar applications”? Explain with examples if it is possible.

Line 505: “Data are shown as mean.” I think it is better “mean ± SD”

Line 507: Change “FSH secretion” for “RBD secretion”.

Line 516: Add “Data are shown as mean ± SD” before N=2.

Line 561: Add “mean ± SD” and plot SD for each sample in the Figure 4E.

There are “Results and discussion” and “Discussion” sections in the manuscript. The authors have to choose if “Discussion” section will be separated from “Results” section.

Reviewer #2: In this manuscript Sinegubova et al describe a system to express the RBD from SARS-Cov2 in CHO cells in large quantities. Many attempts to generate this type of systems have been done recently and this one provide some additional tools to improve the expression of the protein S suitable for immunological tests. The authors use a plasmid vector they generate with the EEF1A 1 promoter and DHFR system to select and amplify to plasmids to generate appreciable expression of the protein. They further characterize the purified protein as a glycosylated monomer and detect reactivity with serum from infected patients. Therefore, the manuscript will be interesting and provide additional tools.

I have some comments that could help to improve the quality of the manuscript

1- The English could be further improved. There are some sentences that could be corrected.

-exemples among others: Line 522 "then?! in the case...""

, all paragraph 540.

2- The introduction, although valuable could be more direct and avoid repetitions.

3- Line 108, explain better the sentence and provide a reference for it.

4- Line 127, provide references for the affirmation.

5- Line 392, what is YP_009724390.1?

6-The authors cite reference 26 as a source of their sequence. They should make it more explicit which one. By adding the plasmids in Addgene and providing GeneBank numbers, they satisfy the exigences to make available their information. Unfortunately, it is not available to the reviewer, so it cannot be checked.

7- Text reorganization

- The discussion about the reasons why initially the protein was eluted with 50 mM Imidazol should be done after figure 2 and the paragraph should state that. It is odd to conclude that a component of the culture medium affect it. Which component it is? Another medium was tried?

- The panels of figure 3 should be arranged as they appear in the text.

8- The cell numbers as nlm should be replaced by exp6

9- The proposal for the internal Cys-Cys bridge is speculative. V2 decrease in apparent size and V1 increase. Why? and why 2 bands appear? Usually loading samples in the same gel causes some degree of diffusion of DTT.

10- Although not relevant for publication in PlosOne, it would be interesting to have some clue if the protein is recognized by neutralizing antibodies and if the attained conformation is homogenous. This could be mentioned in the Discussion.

6. PLOS authors have the option to publish the peer review history of their article (what does this mean?). If published, this will include your full peer review and any attached files.

Reviewer #1: No

Reviewer #2: **Yes: **Sergio Schenkman

---

## [Author Response · Author response to Decision Letter 0]

23 Dec 2020

We would like to thank both Reviewers for their time and effort. 

The article manuscript was updated accordingly to their requests. Reviewers’ comments and responses are as follows:

Reviewer #1 comments

1. The result was good and similar for both versions of RBD. However, there is no mention of perspectives and next steps for the present work by the authors.

I suggest the authors to include a conclusion paragraph at the end of “Discussion” section or add “Conclusion section” to sum up the importance of the results obtained in the manuscript.

Response: We added two paragraphs to the Discussion section, summarizing the findings of the study and describing the possible further research activity:

“The described method for obtaining the 320-537 RBD fragment of the SARS-CoV-2 S-protein can be immediately employed to produce a highly consistent RBD reagent for serological assays. The same cell line creation strategy will be applied to obtain RBD 320-537 protein without C-terminal tag, potentially suitable for subunit vaccine studies. This variant of the RBD protein will be characterized in much more detail, including direct analysis of N-linked and O-linked glycans, peptide mapping with S-S bridges positions quantification and surface plasmon resonance studies of interaction with the ACE2 receptor and known neutralizing antibodies. Highly productive RBD-secreting CHO cell lines are already mentioned in the SARS-CoV-2 RBD dimer vaccine candidate studies but without any details on the cell line generation methodology [42].

Other S-protein domains and multi-domain subunits, suitable for serological assays or vaccine studies, can be obtained in stably transfected CHO cells using the vector plasmid pTM described in this study. Although we found no visible changes in the properties of the RBD 319–541 and RBD 320-537 variants used as ELISA antigens, there is a sharp difference in the antigenic properties of the RBD monomer and dimer, described in [42] – dimeric beta-CoV RBD produced much higher titers of neutralizing antibodies due to unknown reasons. Thus, SARS-CoV-2 RBD preparations with a high content of the covalent dimer admixture, i.e., 319–541 variant, can produce incorrectly positive results in the subunit vaccine studies; this variant should be replaced by the shorter 320-537 RBD variant, containing much lower content of the covalent dimer admixture.”

2. Line 52-53: Explain the last sentence better. RBD produced was tested in ELISA and the authors mention “the described technique is suitable for serological tests and similar applications.” What is the meaning of “similar applications”? Explain with examples if it is possible.

Response: we changed the “similar applications” to the “subunit vaccine studies”; we agree that “serological testing” describes all realistic non-vaccine use of the RBD.

3. Line 505: “Data are shown as mean.” I think it is better “mean ± SD”

Response: corrected

4. Line 507: Change “FSH secretion” for “RBD secretion”.

Response: corrected

5. Line 516: Add “Data are shown as mean ± SD” before N=2.

Response: corrected

6. Line 561: Add “mean ± SD” and plot SD for each sample in the Figure 4E.

Response: line 561 corrected, error bars plotted, in most cases they are too small to be visible, in the wireframe view they look as follows: [illustration 1, may be found in the attached Word file]

7. There are “Results and discussion” and “Discussion” sections in the manuscript. The authors have to choose if “Discussion” section will be separated from “Results” section.

Response: corrected to Results - Discussion

Reviewer #2 (Dr. Sergio Schenkman) comments:

1- The English could be further improved. There are some sentences that could be corrected.

- exemples among others: Line 522 "then?! in the case...""

, all paragraph 540.

Response: Line 522 changed to “At the same time, higher cell productivity for RBDv2 was achieved not due to an increase in gene copy number compared with RBDv1”.

We rewrote paragraph 540 to improve the English language and clarity

2- The introduction, although valuable could be more direct and avoid repetitions.

Response: We made the Introduction section shorter by removing repetitions and information not directly related to the article's topic.

3- Line 108, explain better the sentence and provide a reference for it.

Response: We excluded this phrase. 

4- Line 127, provide references for the affirmation.

Response: Reference 13 (new numbering) describes the use of full S-protein ectodomain for ELISA testing of COVID-19 patients along with other S-protein fragments. We introduced this reference here and renumbered all references accordingly.

5- Line 392, what is YP_009724390.1?

Response: This is the record in the NCBI Protein database https://www.ncbi.nlm.nih.gov/protein/1796318598, S-protein reference sequence. We changed this sentence accordingly.

6-The authors cite reference 26 as a source of their sequence. They should make it more explicit which one. By adding the plasmids in Addgene and providing GeneBank numbers, they satisfy the exigences to make available their information. Unfortunately, it is not available to the reviewer, so it cannot be checked.

Response: We changed the sentence at line 174 to the following one: “p1.1-Tr2-RBDv1 construction. The RBD 319–541 coding sequence was synthesized according to [13], synthetic gene SARS_CoV_2RBD_his [GenBank: MT380724.1].”

We apologize for the delay in Addgene and Genebank data processing pipelines. We have added sequences of all four plasmids in the .gb format as Supporting data. We hope that both Addgene and Genbank will make plasmid data visible soon. We did not request any specific release date for our sequence data. Therefore, our records will be immediately released to the public database once they are processed.

7- Text reorganization

Response: We made many changes to the text and changed the Introduction and Discussion part significantly.

[7.1]- The discussion about the reasons why initially the protein was eluted with 50 mM Imidazol should be done after figure 2 and the paragraph should state that. It is odd to conclude that a component of the culture medium affect it. Which component it is? Another medium was tried?

Response: We added the explanation to the text after Figure 3 – “In the case of Ni-IDA ligand, interaction with the 6xHis tag on the RBDv1 protein was insufficient to retain the target protein on the column in 50 mM imidazole, resulting in premature elution of RBDv1 along with contaminating cell proteins. In the case of Ni-NTA ligand, both 6xHis and 10xHis tags were suitable for efficient removal of proteinaceous admixtures. This observation is in line with the general properties of immobilized metal affinity chromatography resins. [33].”

Unfortunately, proper comparative studies of IDA and NTA ligands are very rare in scientific journals. Simple and sound explanation of the differences between IDA and NTA resins can be found here - https://cube-biotech.com/us/nta-versus-ida-what-s-the-difference, but this web site was not peer-reviewed and cannot be properly cited.

Culture medium incompatibility with the Ni-IDA and Ni-NTA resins is a common topic. For example, it was mentioned in the description of the Expi293 medium by the manufacturer as follows: “Can I purify my protein from Expi293 Expression Medium using ProBond or Ni-NTA purification systems? Answer: Expi293 Expression Medium is not directly compatible with ProBond or Ni-NTA purification systems. We recommend performing a buffer exchange or dialyzing the samples before His-tag purification.” https://www.thermofisher.com/order/catalog/product/A1435103#/A1435103

In some specialized bioprocessing articles, this question was studied in more detail; for example, here - https://www.sciencedirect.com/science/article/pii/S0021967320307809 . The loss of protein binding in the presence of the CHO expression media can be seen as the decrease in the cyan-colored bars' size on Figures 6 B and 7 B, rightmost bar charts, sixfold capacity drop is depicted but not mentioned in the article text directly.

As we aim to purify large quantities of the RBD protein, pre-column ultrafiltration/diafiltration operation will be obligate for volume reduction of the culture supernatant and keeping the sample application time within reasonable time limits (2 h for the complete chromatography operation, from sample application start to the product elution finish). We do not plan to change the culture medium to another, IMAC-compatible, because the stably transfected cell population described in this manuscript might not adapt well to another medium.

[7.2]- The panels of figure 3 should be arranged as they appear in the text.

Response: Corrected, panel letters were rearranged and figure descriptions were changed accordingly.

8- The cell numbers as nlm should be replaced by exp6

Response: Corrected, as well as 105 - to exp5.

9- The proposal for the internal Cys-Cys bridge is speculative. V2 decrease in apparent size and V1 increase. Why? and why 2 bands appear? Usually loading samples in the same gel causes some degree of diffusion of DTT.

Response: We have made 2 different statements, concerning Cys-Cys bridges: 

1) Unpaired C-terminal Cys in RBDv1 makes an intermolecular bridge to another RBDv1 molecule, forming the covalent dimer. 

2) Double-band pattern of both RBDv1 and RBDv2 protein bands in non-reducing conditions may be explained by the formation of two various sets of intramolecular Cys-Cys bridges. 

The second statement is speculative; it was already marked as the proposal, not the proven fact, by the request of Reviewer #1. 

The first statement is based on the simple visual observation – in Figure 4 A, one may see an intense protein band on ~70 kDa for RBDv1, non-reducing conditions. This band completely disappears in reducing conditions. The same band pattern may be seen for RBDv2 protein, although the covalent dimer band intensity is much lower, 6% from total by gel densitometry. Approximately the same level of all multimeric forms in RBDv2 is detected by size-exclusion chromatography. Same observations were made in another work, doi: 10.1101/2020.07.31.231282, cited by us. Possible diffusion of DTT in the case of Figure 4 A was prevented by the introduction of two empty lanes between reduced samples and non-reduced samples; usually, if DTT effectively reaches the non-reducing lane, it will cause a gradient-like shift in bands mobility, and this is not the case on Figure 4 A. Example of such shift is shown here: [illustration 2, may be found in the attached Word file]

We used one gel piece for both reduced and non-reduced samples because relative protein bands mobility on one gel piece may be compared directly. As one may see, both protein variants appear as lower molecular weight proteins in reducing conditions if compared to non-reducing conditions. Another gel shift, caused by the PNGase F treatment (the enzyme itself is seen as the faint 35 kDa band), is not entirely equal for RBDv1 and RBDv2 proteins. This is normal because N-glycan composition for these two protein variants might not be the same – cells with lower specific productivity might produce more heavily processed and branched complex-type glycans so that the resulting glycoprotein molecule would be more bulky. Detailed glycoprofiling of RBD variants is definitely out of the scope of this work; we plan to analyze the tag-free RBD produced by the clonal cell line and intended for vaccine candidate studies. 

Two adjacent bands in reducing conditions can be seen for RBDv1, but not for RBDv2. A possible explanation is that RBDv1 has two major glycoforms with various glycan structures because RBDv1 protein is a pretty thin single band after PNGase F treatment. This explanation is the simplest one; it is not proven by additional experiments, so we do not discuss this matter in the article. RBDv1 protein variant is sub-optimal; we think it should be left behind for many reasons, including this strange band doubling.

 10- Although not relevant for publication in PlosOne, it would be interesting to have some clue if the protein is recognized by neutralizing antibodies and if the attained conformation is homogenous. This could be mentioned in the Discussion.

Response: We added to the Discussion section the following sentence: “This variant of the RBD protein will be characterized in much more detail, including direct analysis of N-linked and O-linked glycans. Peptide mapping with S-S bridges positions quantification and surface plasmon resonance studies of interaction with the ACE2 receptor and known neutralizing antibodies.”

Generally, we do not think that we have any conformational issues, since the 320-537 RBD variant covers entirely the folded RBD domain, as it was resolved by cryo-EM studies and we know, that our protein variant is glycosylated similarly to the 319-541 variant, according to gel shift (Fig. 4A) and MALDI MS data. We plan to characterize the tag-less RBD 320-537 along with the RBDv2 by the Biacore with ACE2, but at present this study is not finished yet.

Sincerely,

Nadezhda A. Orlova, Ph.D.

research scientist

Laboratory of Mammalian Cell Bioengineering, Institute of Bioengineering, Research Center of Biotechnology of the Russian Academy of Sciences. 7, bld. 1, 60 let Oktjabrja pr-t, Moscow 117312, Russia; +7(916)184-4661 nobiol@gmail.com

---

## [Decision Letter · Decision Letter 1]

14 Jan 2021

High-level expression of the monomeric SARS-CoV-2 S protein RBD 320-537 in stably transfected CHO cells by the *EEF1A1*-based plasmid vector

PONE-D-20-36259R1

Dear Dr. Orlova,

We’re pleased to inform you that your manuscript has been judged scientifically suitable for publication and will be formally accepted for publication once it meets all outstanding technical requirements.

Kind regards,

Paulo Lee Ho, Ph.D.

Academic Editor

PLOS ONE

Additional Editor Comments (optional):

Reviewers' comments:

Reviewer's Responses to Questions

**Comments to the Author**

1. If the authors have adequately addressed your comments raised in a previous round of review and you feel that this manuscript is now acceptable for publication, you may indicate that here to bypass the “Comments to the Author” section, enter your conflict of interest statement in the “Confidential to Editor” section, and submit your "Accept" recommendation.

Reviewer #1: All comments have been addressed

Reviewer #2: All comments have been addressed

2. Is the manuscript technically sound, and do the data support the conclusions?

Reviewer #1: Yes

Reviewer #2: Yes

3. Has the statistical analysis been performed appropriately and rigorously? 

Reviewer #1: Yes

Reviewer #2: Yes

4. Have the authors made all data underlying the findings in their manuscript fully available?

Reviewer #1: Yes

Reviewer #2: Yes

5. Is the manuscript presented in an intelligible fashion and written in standard English?

Reviewer #1: Yes

Reviewer #2: Yes

6. Review Comments to the Author

Reviewer #1: The authors have adequately addressed the comments from previous round of review.

I have few considerations for the authors:

- Line 594: Correct “mln”.

- The figure titles and table names presented in “supp-2.docx” file were not changed according to lines 791-817.

Reviewer #2: Thanks for the corrections.

The manuscript is improved and my questions were adequately answered.

7. PLOS authors have the option to publish the peer review history of their article (what does this mean?). If published, this will include your full peer review and any attached files.

Reviewer #1: No

Reviewer #2: **Yes: **Sergio Schenkman

---

## [Editor Report · Acceptance letter]

22 Jan 2021

PONE-D-20-36259R1 

High-level expression of the monomeric SARS-CoV-2 S protein RBD 320-537 in stably transfected CHO cells by the *EEF1A1*-based plasmid vector 

Dear Dr. Orlova:

I'm pleased to inform you that your manuscript has been deemed suitable for publication in PLOS ONE. Congratulations! Your manuscript is now with our production department. 

Kind regards, 

on behalf of

Dr. Paulo Lee Ho 

Academic Editor

PLOS ONE